# Patterning and dynamics of membrane adhesion under hydraulic stress

Céline Dinet [1,8,10], Alejandro Torres-Sánchez [2,3,9,10], Roberta Lanfranco[4], Lorenzo Di Michele[5,6], Marino Arroyo [2,3,7,11] ✉ & Margarita Staykova [1,11] ✉

Hydraulic fracturing plays a major role in cavity formation during embryonic development, when pressurized fluid opens microlumens at cell-cell contacts, which evolve to form a single large lumen. However, the fundamental physical mechanisms behind these processes remain masked by the complexity and specificity of biological systems. Here, we show that adhered lipid vesicles subjected to osmotic stress form hydraulic microlumens similar to those in cells. Combining vesicle experiments with theoretical modelling and numerical simulations, we provide a physical framework for the hydraulic reconfiguration of cell-cell adhesions. We map the conditions for microlumen formation from a pristine adhesion, the emerging dynamical patterns and their subsequent maturation. We demonstrate control of the fracturing process depending on the applied pressure gradients and the type and density of membrane bonds. Our experiments further reveal an unexpected, passive transition of microlumens to closed buds that suggests a physical route to adhesion remodeling by endocytosis.

Hydraulic pressure is a major force at cellular and tissue scales[1,2] compromising tissue integrity[3,4], cell fate decisions[5,6], embryo development[5,7,8], or organ morphogenesis[9]. Some of these processes rely on the ability of hydraulic pressure to selectively detach cell–cell or cell-matrix adhesions. For instance, hydraulic cell–cell fracture is thought to determine the first stages of mammal development, during which pressurization of the gaps at cell–cell junctions produces a widespread distribution of small blisters, which subsequently undergo an actively guided coarsening process to form the blastocoel[7]. Immediately subsequent stages of development involve further luminogenesis and water management between lumens[8]. In vitro studies have shown that cell-autonomous[10], poroelastic[3,11], or osmotically applied[11–13] pressure differences can lead to patterns of pressurized blisters of various sizes, from sub-micron cracks to multicellular cavities.

The formation and early dynamics of hydraulically generated blisters should be a largely physical process, which cells and tissues may tune by regulating physical parameters in space and time, and whose guidance determines the subsequent shape of organs (network of bile ducts), the resilience of the epithelial barrier under hydraulic stress or stretch[3] or the robustness of morphogenesis[7]. While the coarsening of an array of preexisting microlumens has been studied[7,14], the physical principles controlling when and how hydraulic cracks emerge in the first place from a pristine adhesion remain largely unknown.

[1]Department of Physics, Durham University, Durham, UK. [2]Universitat Politècnica de Catalunya-BarcelonaTech, 08034 Barcelona, Spain. [3]Institute for Bioengineering of Catalonia (IBEC), The Barcelona Institute of Science and Technology, 08028 Barcelona, Spain. [4]Department of Physics, Cavendish Laboratory, University of Cambridge, Cambridge, UK. [5]Department of Chemical Engineering and Biotechnology, University of Cambridge, Cambridge, UK. [6]Department of Chemistry, Imperial College of London, London, UK. [7]Centre Internacional de Mètodes Numèrics en Enginyeria (CIMNE), 08034 Barcelona, Spain. [8]Present address: Laboratoire de Chimie Bactérienne, Institut de Microbiologie de la Méditerranée, CNRS-Aix-Marseille University, 31 Chemin Joseph Aiguier, 13009 Marseille, France. [9]Present address: European Molecular Biology Laboratory (EMBL-Barcelona), 08003 Barcelona, Spain. [10]These authors contributed equally: Céline Dinet, Alejandro Torres-Sánchez. [11]These authors jointly supervised this work: Marino Arroyo, Margarita Staykova. ✉e-mail: marino.arroyo@upc.edu; margarita.staykova@durham.ac.uk

To identify these principles, in this work we combine experimental observations of adhered lipid vesicles with mathematical and computational modeling. The hydraulic fracture in embryonic tissues is driven by pressure gradients established through active ion transport across cell membranes, followed by a passive compensatory efflux of water into the cell–cell interstice[7]. To generate such pressure gradients in our artificial system, we subject the vesicles to osmotic shocks and observe hydraulic fracturing dynamics akin to those in cells. Our approach allows us to access and control the various mechanical and transport mechanisms controlling the formation of membrane hydraulic cracks, their emerging spatial patterns and characteristic coarsening dynamics.

## Results

### Hydraulic fracturing in lipid vesicles

In our default experiments, we use giant lipid vesicles (GUV) adhered to supported lipid bilayers (SLB) or to other GUVs via biotin (b)–neutravidin (NAV) bonds, chosen to mimic the mobile linkers between cells (Fig. 1a–c). The density of b-NAV bonds is controlled by the molar ratio of biotinylated lipids in the membranes ("Methods"). Such systems have been previously used to understand adhesion between fluid membranes[15,16]. At low linker densities, the lipids and

bonds in the adhesion zone between vesicles are rapidly diffusing, with slower diffusion observed for supported GUV-SLB systems due to substrate-related friction and pinning effects (Supplementary Fig. 1). At higher bond density, the lateral crowding of the NAVs confined within the thin (5–6 nm) interstitial space restricts the lipid and NAV mobility[15,16], and at 4 mol%—the highest density of biotinylated lipids we reach, NAVs are close to their saturation density[15] and become practically immobile according to Fluorescent Recovery after Photobleaching (FRAP) experiments (Supplementary Fig. 1c, d).

The adhered vesicles in equilibrium have the shape of a truncated sphere (Fig. 1a) with sharp contact angles and no noticeable membrane fluctuations, indicative of a strong adhesion regime. The adhesion patch exhibits homogeneous NAV fluorescent intensity, which is higher than the surrounding SLB (Fig. 1a and Supplementary Fig. 2), in agreement with previously observed enrichment of bonds between fluid membranes[15,16]. When we subject the vesicles to hyper-osmotic shock, applied by rapidly increasing the concentration of osmolytes in the external vesicle medium, they deflate and increase their adhesion footprint (for example the vesicle on Fig. 1a loses about 20% of its volume, see Supplementary Note 1). Most interesting, however, we observe fracturing of the membrane adhesion contacts with the for-

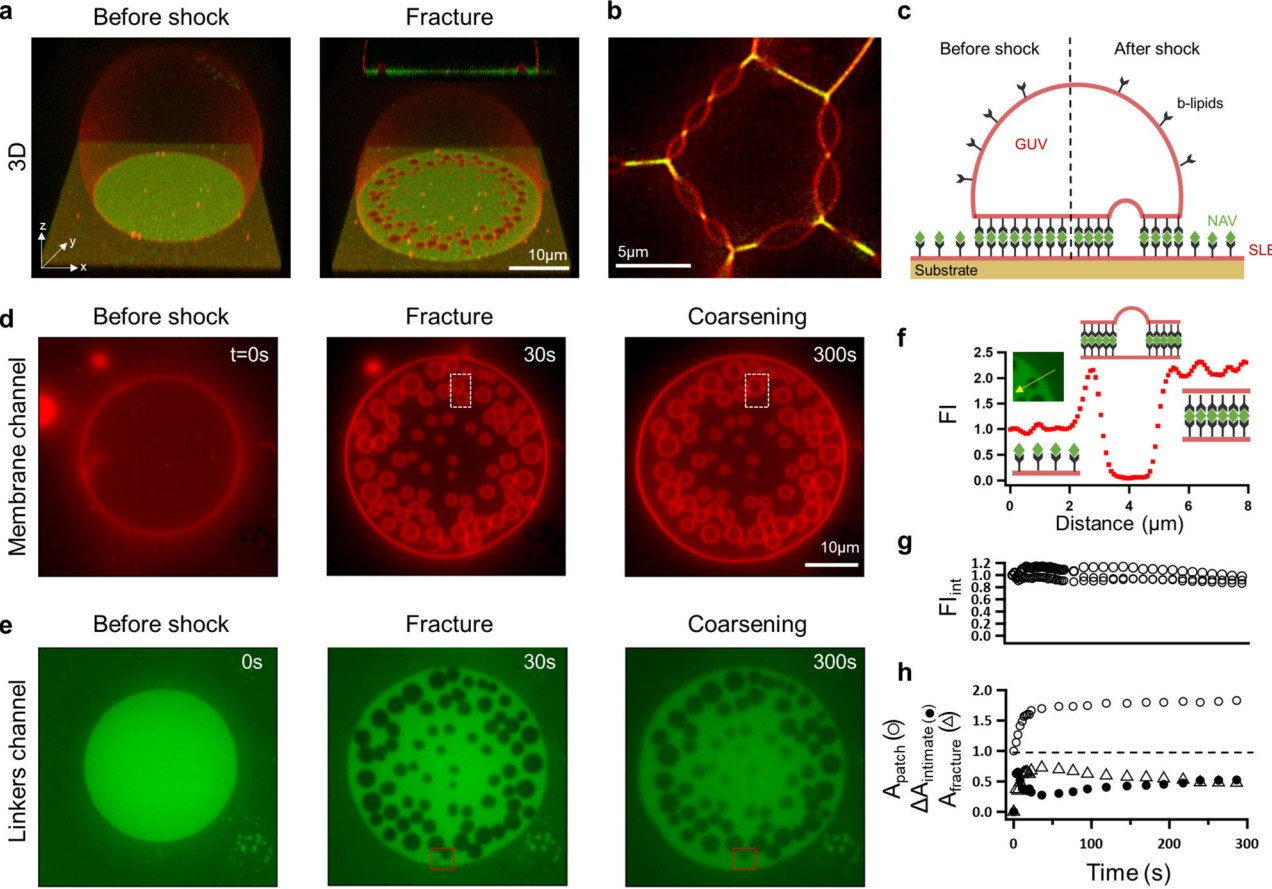

**Fig. 1 | Hydraulic fracturing of adhered lipid membranes. a** 3D images of a vesicle bound to a supported membrane (at 1 mol% biotinylated lipids) before (0 s) and after (60 s) the application of a hyper-osmotic shock of 100 mM; inset represents *xz* cross-section of the vesicle after the shock. The membrane, labeled by Rhodamine, appears in red and the NAV bonds, labeled by DyLight488, in green. **b** Blister formation at GUV-GUV interfaces at 0.5 mol% biotin density and 100 mM osmotic shock. **c** Sketch of membrane and bond distribution, before and after the shock. **d, e** Images of the GUV-SLB adhesion zone (0.2 mol% biotin density, 100 mM osmotic shock) displaying the lipid membrane (**d**) and the NAV distribution (**e**) at 0, 30 and 300 s after the shock. The dashed squares show blister fusion (white) and

blister collapse (red) events. **f** NAV fluorescent intensity profile across a single blister, FI, normalized to the NAV intensity in the SLB, along the yellow arrow in the inset. **g** NAV fluorescent intensity integrated over the area of intimate membrane adhesion (FI$_{int}$) vs time. The plot shows data sets from three different vesicles. **h** Area analyses for the vesicle on (**d, e**): Adhesion patch area ($A_{patch}$), normalized to the initial patch area ($A_{patch}(0)$) as a function of time (open circles); total fracture area ($A_{fracture}$), obtained by combining the dark (NAV-devoid) footprints of all blisters, and change in intimate adhesion area ($\Delta A_{intimate} = A_{intimate} - A_{patch}(0)$) over time, both as a fraction of the total patch area spreading, $A_{patch} - A_{patch}(0)$ (triangles and closed circles, respectively).

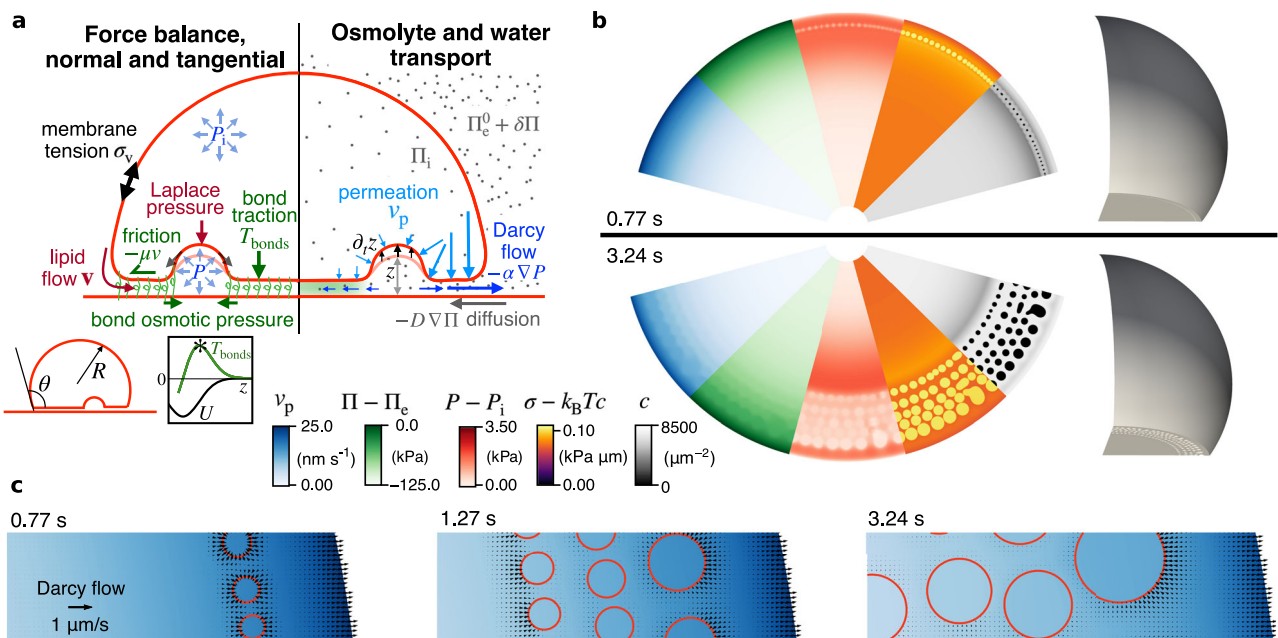

**Fig. 2 | Mechanism of hydraulic fracturing. a** Schematic of the physical ingredients controlling the formation of a pattern of hydraulic cracks. Mechanical force balance (left) and osmolyte and water transport (right) allow us to solve for hydraulic and osmotic pressures inside the vesicle $P_i(t)$ and $\Pi_i(t)$ and in the interstice $P(x, y, t)$ and $\Pi(x, y, t)$, for the membrane velocity $\mathbf{v}(x, y, t)$, for bare membrane tension, for bond concentration and for the vesicle shape given by $R(t)$, $\theta(t)$ (left inset) and height $z(x, y, t)$. The right inset shows the non-monotonic relation between bond traction $T_{bonds}$ and $z$, along with the effective adhesion potential $U(z)$. Other symbols are described in the text and Supplementary Note 2. **b** Snapshots of various fields at the adhesion patch and 3D view of the membrane. The blue map is permeation velocity, proportional to the water chemical potential across the membrane, $P(x, y, t) - \Pi(x, y, t) - (P_i - \Pi_i)$. The green map is the osmotic pressure relative to that of the external medium $\Pi(x, y, t) - \Pi_e$. The red map is the hydraulic pressure relative to the vesicle pressure, $P(x, y, t) - P_i$, whose gradient is proportional to Darcy flow. The purple map is the membrane tension, $\sigma - k_B T c$, and the gray map is the bond concentration. **c** Zoom of water transport in the interstice, with water permeation (color map) and Darcy flow (arrows); pockets are outlined in red color. Model parameters are reported in Supplementary Table 1.

mation of multiple blisters (Supplementary Movie 1). In the GUV-SLB system, the blisters look like spherical caps protruding into the vesicle (Fig. 1a), whereas in GUV-GUV adhesions, we observe lenticular blisters (Fig. 1b) remarkably similar to those in early mammalian embryos[7]. The size and distribution of blisters appear to vary with linker density and osmotic shock (Supplementary Fig. 3). Common for all blisters is the lack of NAV signal in the separated membranes (Fig. 1a, b, e), which implies that bonds do not break but are laterally displaced during blister formation (Fig. 1c, f). This is also the case for the systems with the highest linker density, despite their limited FRAP mobility (Supplementary Fig. 1). Bond displacement is also confirmed by the fact that the NAV density integrated over the area of intimate membrane contact (FI$_{int}$) remains constant during the fracturing process, i.e. the total number of bonds remains constant (Fig. 1g and Supplementary Fig. 4).

In strongly adhered vesicles, the excess membrane area following deflation is expected to lead to further vesicle spreading[17,18]. Even though we observe an expansion of the total adhesion patch area, $A_{patch}$ (Fig. 1a, h), a closer analysis reveals that it is mostly driven by opening of blisters, quantified as $A_{fracture}$, rather than from an expansion of intimate membrane adhesion (Fig. 1h, $A_{fracture} > \Delta A_{intimate}$). This behavior is conserved for different shock magnitudes and linker densities (Supplementary Fig. 4). Following the blistering process, there is a gradual decrease in $A_{fracture}$ accompanied by gradual increase in the intimate adhesion area, $\Delta A_{intimate}$, due to either coalescence of blisters or blister collapse (Fig. 1h). The latter has been interpreted as a process akin to Ostwald ripening during which the collapsing blister transfers its content to neighboring ones by diffusive water transport in the tightly adhered interstice[7] (Fig. 1d, e, Supplementary Movie 1).

## Theoretical model

To understand the experimental observations of hydraulic fracturing, we develop a mathematical model predicting the nucleation and

evolution of hydraulic blisters following an osmotic perturbation from a pristine adhesion patch. The model self-consistently couples transport and mechanical phenomena in the adhesion patch and in the free-standing vesicle (Fig. 2a). Briefly, we align the $(x, y)$ plane with that of the SLB and assume fast equilibration of osmotic and hydraulic pressures in the interstice along the thin $z$ direction. Lateral transport of osmolytes is controlled by diffusion (with coefficient $D$) and advection of the aqueous solution. Transport of water in the adhesion cleft includes water permeation across the membrane (with permeability $K$) and lateral flows, which by approximating the thin and crowded interstice as a porous medium, we assume to be proportional to gradients of hydraulic pressure following Darcy's law with mobility $\alpha$[19]. Because water is incompressible, the imbalance between convergent/divergent lateral Darcy flows and water permeation across the membrane specifies locally the rate of change of membrane height $z$. The lack of prominent gradients in NAV intensity in the zone of intimate contact following blister formation (Fig. 1a, e) suggests fast lateral equilibration of bonds. Accordingly, we assume that the number density of bonds $c(x, y, t)$ follows a Boltzmann distribution accounting for the interplay between mixing entropy, tending to uniformize bond distribution, and bond stretching, which strongly disfavors the presence of bonds where membrane height $z(x, y, t)$ deviates from the equilibrium separation[20,21].

Turning to force balance, the fluid membrane supports a tangential stress tensor that includes bare tension $\sigma$ ($\sigma_v$ in the free-standing part), the adhesion tension $\gamma$, and tensions induced by membrane shear-rate with viscosity $\eta$. For mobile bonds, $\gamma$ is the 2D osmotic pressure of these molecules trapped within the adhesion rim[20,22], which in a dilute approximation follows $\gamma = k_B T c$ where $k_B$ is Boltzmann's constant, and $T$ the absolute temperature. Normal to the membrane, the membrane stress generates a Laplace pressure, balanced by the difference of hydraulic pressures across the membrane and by the traction due to bond stretching $T_{bonds}$ keeping the

membranes together (Fig. 2a). Tangential gradients in membrane stress generate membrane flows balanced by friction (with coefficient $\mu$), whose physical origin is the resistance to lipid flow posed by an array of membrane obstacles, here the bonds[23].

The equations of water, lipid, bond, and osmolyte mass balance, along with membrane mechanical equilibrium, allow us to calculate the time evolution of vesicle variables (shape, tension $\sigma_v$ and internal osmotic and hydraulic pressures, $\Pi_i$ and $P_i$) and of interstitial fields (membrane height $z(x, y, t)$, osmotic and hydraulic pressures $\Pi(x, y, t)$ and $P(x, y, t)$, membrane tangential velocity $\mathbf{v}(x, y, t)$, bare membrane tension $\sigma(x, y, t)$ and bond density $c(x, y, t)$), Fig. 2a.

We provide in the Theory Box a self-standing and complete summary of the unknowns, governing equations, initial and boundary conditions, and in Supplementary Note 2 a derivation of the model equations from balance principles and constitutive relations. We further develop a finite element method[24,25] to solve the model above in its full nonlinearity for the default GUV-SLB system. The parameter values are derived from our experiments and from the literature (Supplementary Table 1 and Supplementary Note 3).

## Mechanisms of hydraulic fracturing
Our numerical simulations readily develop arrays of hydraulic blisters upon osmotic shock application, which closely resemble our experimental observations (Fig. 2b, Supplementary Movie 2), and at the same time provide us with access to all physical fields with high temporal and spatial resolutions. This allows us to examine in detail the initial stages of blister formation. Right after the shock, osmolyte diffusion from the outer medium into the interstice rapidly increases the osmotic pressure close to the rim of the adhesion patch (green map in Fig. 2b). Without time to change $z(x, y, t)$ significantly, mechanical equilibrium normal to the membrane implies that the hydraulic pressure $P(x, y, t)$ stays nearly constant and hence the local increase of $\Pi(x, y, t)$ is mirrored by a decrease of water chemical potential across the membrane ($P - \Pi - (P_i - \Pi_i)$) in the margin, which drives permeation efflux into the interstitial region (blue map in Fig. 2b, c). Very close to the edge, water permeating from the deflating vesicle into the interstice can escape to the outer medium (black arrows in Fig. 2c). However, at a distance from it, water is hydraulically confined by Darcy resistance and accumulates, thus increasing locally $z$.

The local swelling of the interstice leads to loss of membrane adhesion even though bonds are not dissociated, as our experiments show. According to our theory, bond motion is driven by gradients of entropic and stretching chemical potentials, the latter of which strongly push bonds away from regions where membranes are drawn apart. The gradients in stretching chemical potential during blister formation are much larger than those of entropic origin generated in our FRAP experiments (Supplementary Fig. 1), hence explaining why bonds with reduced mobility at high concentrations appear immobilized during FRAP, but mobile during fracturing. This phenomenology is consistent with a microscopic picture of biased Brownian motion with stick-slip nonlinear friction[26,27]. Adhesion patches with high bond density behave similarly to dynamically arrested dense colloidal glasses, which attain fluid-like behavior only beyond a certain yield stress[28,29]. In the process of blister formation, membrane separation not only decreases locally bond concentration, but also increases the force born by individual bonds remaining in this region, leading to a non-monotonic traction-separation relation, $T_{bonds}(z)$. The traction reaches a maximum $T^*_{bonds} = \sqrt{k_B T k c}$ at critical separation $z^* = z_0 + \sqrt{k_B T / k}$, where $k$ is the bond stiffness[20,21], and then decreases to zero supporting a dissociated state (Fig. 2a, inset). Accordingly, if the osmotic shock $\delta\Pi$ is large enough compared to $T^*_{bonds}$, the uniform adhesion looses stability and partitions into separated bond-free and tightly adhered bond-rich phases by bond rearrangement.

The initial adhesion instability occurs at a certain distance from the edge, determined from the interplay between permeation and

hydraulic confinement (gray map in Fig. 2b). The theoretical model predicts initially axisymmetric ring-like fracture, which rapidly splits into droplet-like spherical blisters as a result of a symmetry-breaking transition akin to a Rayleigh-Plateau instability (Supplementary Movie 2 and Supplementary Fig. 5). The formation of blisters locally relaxes hydraulic pressure (red map in Fig. 2b), and the resulting gradients drive interstitial water flow (Fig. 2c and Supplementary Movie 3). Alongside the persistent water drainage towards the external medium at the edge of the adhesion patch, there are local and transient flows towards growing blisters or during blister coarsening, all of which decay in magnitude over time as the system approaches equilibrium. Analogously, gradients in bare membrane tension drive global lipid flows from the free-standing vesicle into the adhesion patch as blisters form, and local flows are required during blister fusion, growth and collapse (Supplementary Movie 3).

The nucleation and growth of the first row of blisters reduces hydraulic pressure locally, and hence this blister front effectively constitutes a new edge of a smaller pristine adhesion patch (red map in Fig. 2b). The process then repeats with the nucleation and evolution of subsequent rows of blisters (Fig. 2b, Supplementary Movie 2), as long as the vesicle is sufficiently out of osmotic equilibrium to tear the adhesion apart. Our simulations exhibit profuse coalescence as well as events of blister collapse, further discussed later.

## Principles of pattern selection
In the following, we explore how varying the model parameters leads to a wide diversity of fracture patterns in terms of localization, size, spacing, and dynamics. To systematically parse these behaviors, we identify the main non-dimensional numbers controlling the process.

Given a strong enough osmotic perturbation $\delta\bar{\Pi} = \delta\Pi/(\sqrt{k_B T k}c_0) > 1$ to challenge adhesion stability, blister formation additionally requires that the osmotic perturbation penetrates the interstice fast enough compared to the time of overall vesicle osmotic relaxation by permeation $\tau_{osm} = R_0/(K\delta\Pi)$, where $R_0$ is the typical radius of the patch. This effect can be quantified by comparing $R_0$ with the distance of diffusive osmolyte transport during $\tau_{osm}$ given by $\ell_{diff} = \sqrt{D\tau_{osm}}$, resulting in the dimensionless number $\bar{\ell}_{diff} = \ell_{diff}/R_0 = \sqrt{D/(R_0 K\delta\Pi)}$. In agreement with this rationale, our simulations show that if this number is large, then blisters form throughout the adhesion patch. On the contrary, if $\bar{\ell}_{diff}$ is small, blisters only form in a small region close to the edge or do not form at all (Fig. 3a). More quantitatively, we find a linear relation between $\bar{\ell}_{diff}$ and the blister penetration distance normalized by $R_0$ (Fig. 3a, inset).

The condition that $\bar{\ell}_{diff}$ is large enough guarantees significant water permeation into the interstice, and hence is required for pocket formation but not sufficient. For this efflux to pressurize the interstice, it should be opposed by hydraulic resistance, which can be quantified by the dimensionless hydraulic screening length $\bar{\ell}_{scr} = \sqrt{\alpha z_0/K}/R_0$[19]. At a distance smaller than $\bar{\ell}_{scr}R_0$ from the adhesion edge, water can easily leave the interstice by Darcy flow. Thus, if $\bar{\ell}_{scr}$ is comparable or larger than 1, we expect very low hydraulic confinement and no pocket formation. For smaller values, we expect $\bar{\ell}_{scr}R_0$ to determine the distance between the first row of pockets and the edge, as well as the separation between subsequent rows of pockets. Furthermore, since the dynamics of pocket growth requires water volume reconfigurations by permeation and Darcy water flows (Fig. 2c, Supplementary Movie 3), we expect $\bar{\ell}_{scr}R_0$ to dictate the typical size of pockets. These arguments are confirmed by our simulations, including a quantitative linear relation between the normalized distance to the edge of the first pockets $d_{edge}/R_0$ and $\bar{\ell}_{scr}$ (Fig. 3b).

Finally, blister formation out of nearly inextensible lipid membranes requires recruitment of membrane area from the free-standing part of the vesicle and blister reorganization from nearby regions (Fig. 2d). When the dimensionless hydrodynamic length $\bar{\ell}_{hydr} = \sqrt{\eta/\mu}/R_0$ is $\ll 1$ ($\sim 10^{-4}$ in our system), the dominant mechanism

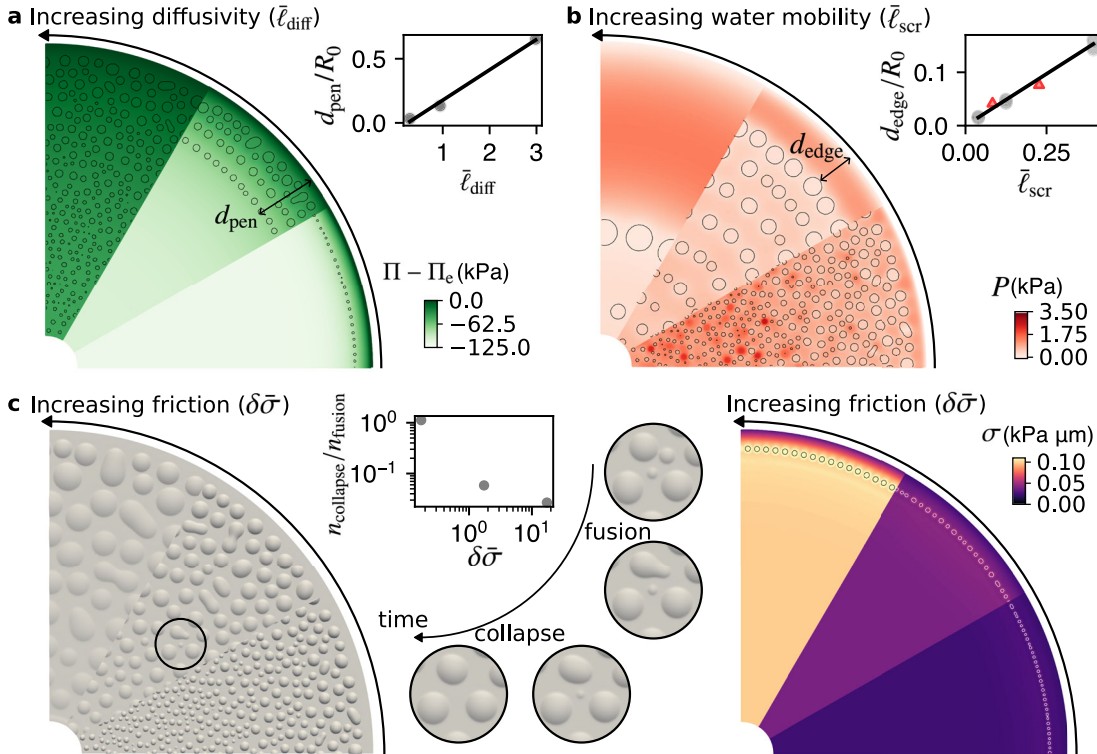

**Fig. 3 | Pattern selection. a** Osmotic pressure relative to the external medium for three different values of $\bar{\ell}_{\mathrm{diff}}$ (0.30, 0.94, 2.98) obtained by changing diffusivity $D$. Pocket boundaries are marked in black. The inset shows the distance between the edge of the patch and the innermost pocket $d_{\mathrm{pen}}$ normalized by patch radius as a function of $\bar{\ell}_{\mathrm{diff}}$. **b** Hydraulic pressure relative to external medium for three different values of $\bar{\ell}_{\mathrm{scr}}$ (0.04, 0.12, 0.39) obtained by changing Darcy mobility $\alpha$. The inset shows the distance between the outermost pocket and the edge $d_{\mathrm{edge}}$ as a function of $\bar{\ell}_{\mathrm{scr}}$. Circles mark simulations where $\alpha$ is changed and triangles where $K$ is changed. **c** Top view of 3D shape of the pockets for three different values of $\delta\bar{\sigma}$ (0.17, 1.73, 17.32) obtained by changing friction $\mu$. The inset shows the relative number of collapses and fusions during pattern coarsening as a function of $\delta\bar{\sigma}$. Zoom images show sequence where both fusion and collapse take place. The map on the right shows bare membrane tension $\sigma$ right after nucleation, showing higher friction generates larger tension gradients. Model parameters are those of Supplementary Table 1 except $D$, $\alpha$, and $\mu$ as noted above, and $\theta_0 = 90°$.

dragging membrane flow and tension equilibration is friction[23]. We can then estimate the lack of bare tension equilibration during pattern formation $\delta\sigma$ produced by frictional delay of membrane recruitment (Supplementary Note 2), leading to a dimensionless number $\delta\bar{\sigma} = \mu\delta\Pi\sqrt{\alpha z_0 K}/(k_B T c_0)$. In agreement with these arguments, our simulations show that for small $\delta\bar{\sigma}$, gradients in $\sigma$ are small, and since $\gamma = k_B T c$ is nearly uniform in the intimate adhesion, all pockets exhibit similar contact angles (Fig. 3c). In contrast, for large $\delta\bar{\sigma}$, our simulations show increasing tension gradients towards the interior of the patch, which are mirrored by a gradient in contact angles. Moreover at large tension, the blisters appear large and shallow (Fig. 3c) as predicted for membrane delaminations that enclose interstitial volume with limited excess membrane[13]. Experimentally, we do not observe shallow blisters with varying contact angles, which implies that our system operates in a low friction regime.

Regarding the coarsening mechanism, our simulations exhibit coexistence of pocket fusion and collapse (Fig. 3c). For low $\delta\bar{\sigma}$, the number of collapse events is similar to the number of fusion events, whereas large $\delta\bar{\sigma}$ favors fusion as blisters grow by laterally expanding their footprint area. Compared to experiments, a significant difference is that in our simulations nearby pockets readily fuse, whereas observations of stable pairs of pockets at very close distance are common (Figs. 1d, e and 4). We attribute the barrier to pocket fusion to the presence of trapped adhesion molecules between pockets.

### Experimental control of fracture patterns

In the light of our model, we now examine how osmotic pressure and membrane-adhesion parameters regulate the hydraulic fracturing. The strength of the osmotic shock, $\delta\Pi$, strongly correlates with the extent

of membrane fracture area, as predicted by theory and simulations. At small shocks, $\delta\Pi = 25$ mM, nearly no blisters are formed, whereas at high shocks $A_{\mathrm{fracture}}$ may account for 50% of the adhesion patch area (Fig. 4a). The increase in $A_{\mathrm{fracture}}$, however, does not take place throughout the whole patch but remains largely confined to the periphery, consistent with the fact that the time for osmolyte penetration and hence $\bar{\ell}_{\mathrm{diff}}$ decrease with increasing $\delta\Pi$.

The density of bonds, which in our experiments is controlled by the fraction of biotynilated lipids, does not only change the initial adhesion strength, but can also affect water and osmolyte transport in the interstitial space by increasing the volume fraction of obstacles[30,31] (Supplementary Note 3). Consistent with this we observe that as bond density increases, the hydraulic fracture predominantly localizes in the external periphery of the membrane adhesion patch (Fig. 4b-i), in agreement with our simulations for small osmolyte diffusivity, $D$ and $\bar{\ell}_{\mathrm{diff}}$. Furthermore, the blisters become smaller, closer to the edge and to each other (Fig. 4b-ii), in agreement with smaller Darcy mobility and decreased $\bar{\ell}_{\mathrm{scr}}$.

While the results above show that bond density controls fracture patterns by osmolyte diffusivity and water mobility, we also expect that the pattern formation would depend on the size and length of the molecular bond. To demonstrate this, we use double-stranded DNA constructs with contour length of about 11.8 nm, which on one side connect to the lipid membrane by a double cholesterol linker, and on the other display a biotin group (Fig. 4c, Supplementary Fig. 6).

The DNA spacers increase the total membrane separation around 5-fold, which reduces the volume fraction of the linker complexes in the interstitial gap (Supplementary Note 3). As a result, even for the highest density of DNA-b-NAV bonds ("Methods"), blisters appear

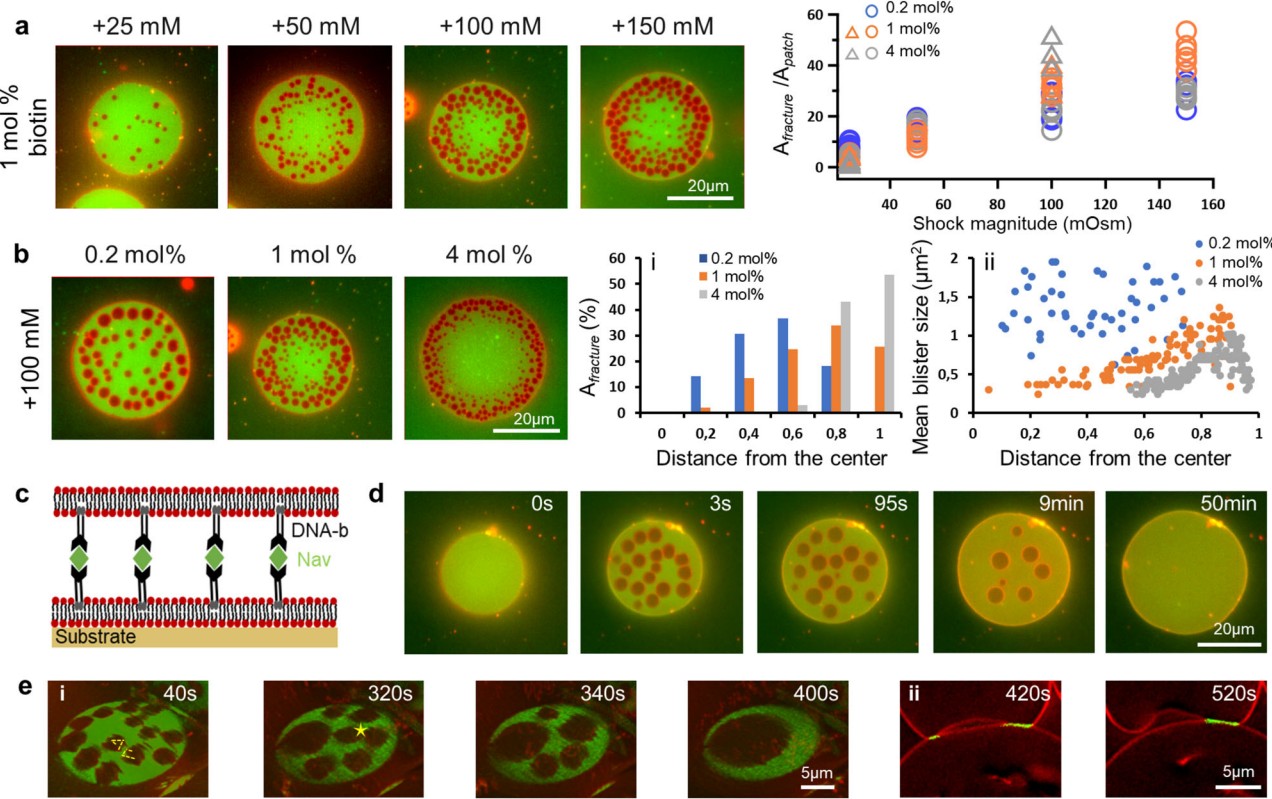

**Fig. 4 | Hydraulic fracture patterns for various experimental parameters.**
**a** Effects of osmotic pressure. Left: Epi-fluorescent images of the GUV-SLB adhesion zone for 1 mol% biotinylated lipids showing hydraulic fracturing at hyper-osmotic shocks of 25, 50, 100, and 150 mM. Plot shows the normalized fractured area as a function of the shock magnitude. Round circles and triangles mark experiments with b-NAV and DNA-b-NAV bonds, and different colors correspond to different linker densities. For each experimental condition we plot data from at least 3 independent experiments (repeating symbols). **b** Effects of bond density. Epi-fluorescent images of adhered vesicles with 0.2, 1, and 4 mol% biotinylated lipids subject to 100 mM hyper-osmotic shock, selected and quantified when all blisters have formed and before they start coarsening, i.e. when $A_{fracture}$ is maximum. For every image, we have quantified the (i) percentage of fracture area, $A_{fracture}$(%) and

(ii) average blister size, as a function of distance from the patch centre. The fractured area in (i) is plotted in 0.2 radius fraction intervals, whereas (ii) shows the size and position of all individual blisters from the microscopic images. **c** Sketch of the DNA systems. The b-NAV bonds are lifted above the membrane surface using double-stranded DNA constructs, linked to the membrane via a double cholesterol anchor (Fig. S1). **d** Time-lapse epi-fluorescent images of adhered vesicles with DNA bond spacers at 6 mol% bond density, subject to 100 mM osmotic shock. **e** GUV-GUV adhesion patch with 0.5 mol% biotinylated lipids after 50 mM osmotic shock; (i) 3D reconstitution of the adhesion zone showing Brownian motion and coalescence of blisters (star). The dashed line shows the displacement of a blister sampled with 20 s time intervals; (ii) Cross-section images of the same adhesion zone at later stages showing blister discharge.

large and uniformly distributed (Fig. 4d, Supplementary Movie 4), in sharp contrast to the peripheral fracture pattern in the b-NAV systems at 4 mol%, and in agreement with larger $\ell_{diff}$ and $\ell_{scr}$. Furthermore, all DNA samples demonstrate enhanced Ostwald ripening, manifested by steady and faster decrease of fractured area (Supplementary Fig. 7), consistent with larger Darcy water mobility. To test whether our theoretical model agrees with these observations, we perform a simulation with increased linker length as well as increased osmolyte diffusivity and Darcy mobility according to the smaller volume fraction of DNA linker complexes in the interstice (Supplementary Note 3). In agreement with experiments, our simulation of the DNA system develops larger and uniformly distributed blisters within one second, which coarsen by enhanced Oswald ripening (Supplementary Movie 5).

The water trapped in the blisters eventually diffuses out, leading to blister collapse, adhesion spreading, and the recovery of a pristine adhesion patch with uniform bond distribution (Fig. 4d, Supplementary Movie 4). This behavior is reminiscent to the healing of hydraulically fractured tissues after the discharge of blisters[3,7], even though it is not driven by an active increase in tension, but by the progressive relaxation of the osmotic pressure difference as water exits the vesicle. By considering smaller system sizes, we can extend the simulations in time to computationally span from the millisecond dynamics of pocket fusion to the tens of minutes required for full equilibration of the

system, reproducing the experimentally observed complete adhesion recovery (Supplementary Movie 6).

Finally, we note that hydraulic fracture patterns appear only when the mobility of bonds and lipids, and hence blisters is significantly reduced. In our experiments, this is the case for all supported GUV-SLB systems, including the DNA samples (Supplementary Fig. 1). In contrast, in GUV-GUV adhesions at low bond density, where pinning and crowding effects are absent, blisters undergo pronounced Brownian motion (Supplementary Movie 7), in close analogy with membrane lipid domains in free-standing membranes vs SLBs[32,33]. Blister mobility masks any initial hydraulic patterning, and also facilitates rapid coarsening by blister coalescence (Fig. 4e-i) and direct discharge to the outer medium (Fig. 4e-ii).

## Budding of blisters
Contrasting with the reversible dynamics leading to healing, we identify an unexpected mechanism leading to irreversible blister budding and bond breaking in most of our b-NAV systems. Over time-scales of several to tens of minutes, blisters undergo a transition from caps to spherical buds in the GUV-SLB patches (Fig. 5a), and from lenticular pockets at the GUV-GUV contacts to pairs of buds (Fig. 5b, Supplementary Movie 8). Buds remain adjacent to the underlying membranes, possibly connected by a narrow neck.

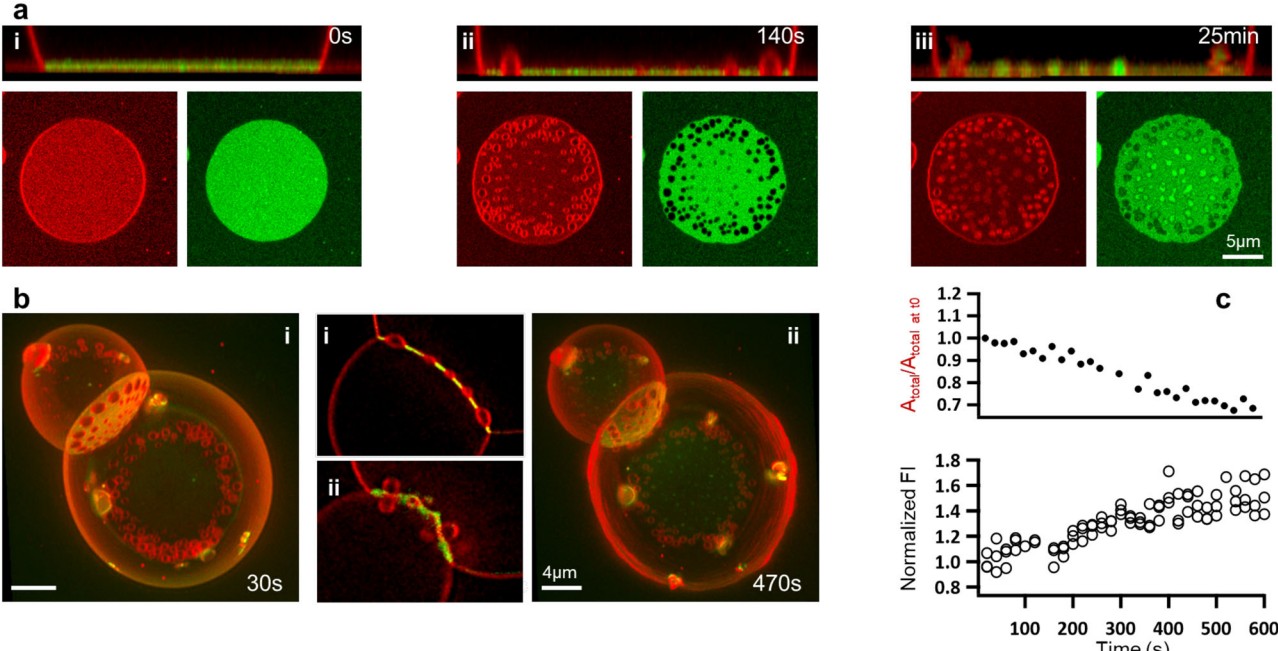

**Fig. 5 | Budding of blisters. a** Confocal images of a GUV-SLB adhesion patch (*xy* and *xz*) at 4 mol% biotinylated lipids, at *t* = 0 (i), 140 s (ii), and 25 min (iii) following a 100 mM osmotic shock. **b** 3D confocal images of two adhered vesicles at 4 mol% biotinylated lipids and subjected to 100 mM osmotic shock at 30 (i) and 470 (ii) s after the shock. The insets show corresponding 2D cross-sections of the GUV-GUV adhesion patch at 0 (i) and 470 (ii) s. **c** Change in adhesion area and linker density (quantified as mean NAV fluorescent intensity in the adhesion zone) over time for two adhered vesicles at 4 mol% biotinylated lipids, subjected to 100 mM osmotic shock. The values in both plots are normalized to the initial values, where *t* = 0 is the time when imaging starts (approximately 30 s after the osmotic shock). The mean NAV fluorescent intensity is corrected for photobleaching and the data points correspond to 3 different *z* planes for each time frame.

To understand these transformations, we first notice that blister-like delaminations in substrate-adhered lipid bilayers transform into buds as a result of increasing excess area or lower membrane tension[13]. Consistent with this notion, we systematically observe pronounced vesicle fluctuations preceding budding transitions (Fig. 5b-ii). Yet, it is not obvious why tension should lower in strongly adhered membrane systems, such as ours. To explain this, we refer to a physical scenario according to which a high- to low-tension transition of strongly adhered vesicles can be the result of hindered spreading[34]. The rationale is that, if during deflation the excess area cannot become adhered by restricted expansion of the contact line, this area will remain in the unattached part, which will become slack and prone to fluctuations. In our system, expansion of the contact line requires motion of bonds. Right after the osmotic shock, fracturing forces can move bonds. However, as the system approaches osmotic equilibrium, bonds become dynamically arrested (as previously discussed regarding FRAP results) and impede further spreading even though the vesicles continue to deflate minutes after the shock. Consistent with this scenario, the patch area remains nearly constant at long times in our b-NAV systems (Fig. 1h and Supplementary Fig. 4) and we observe distinct pinning points and loss of circularity (Fig. 5a-ii). Hence, we posit that the arrested mobility of bonds obstructs vesicle spreading, which in turn reduces tension, enables fluctuations, and causes membrane internalization by cap-to-bud transitions. In contrast, in systems with DNA spacers where the bulky NAVs are drawn apart from the membranes, the larger molecular mobility (Supplementary Fig. 1e, f) allows membrane adhesions to spread avoiding membrane budding and enabling healing of the adhesion patch as shown in Fig. 4d.

The long-term behavior of intimate membrane contacts, accompanying budding is equally intriguing. GUV-SLB patches exhibit loss of adhesion and absence of NAV signal in the vicinity of blisters and the patch boundary (Fig. 5a-iii). In contrast, the GUV-GUV adhesions, shrink in size and intimate area, thus laterally crowding bonds and increasing the NAV signal (Fig. 5c, Supplementary Movie 8). These transformations can be explained by the strong effects of membrane fluctuations on adhesion, which in our systems are further enhanced by the presence of fluctuating blisters. Besides producing an entropic repulsive pressure between membranes[35], fluctuations dynamically pull on bonds increasing their chemical potential[20]. If relaxation by lateral motion is restricted, as in the substrate-supported GUV-SLB systems, the effective affinity of adhesion molecules is reduced favoring bond breaking[21,36–38]. In contrast, GUV-GUV adhesions seem to avoid bond breaking by lateral bond rearrangement[36].

In summary, our experiments demonstrate a transition from strong to weak adhesion mediated by arrested vesicle spreading and concomitant membrane fluctuations, which can break or rearrange bonds[21,36]. More importantly, the reduction of membrane tension by arrested spreading can trigger an irreversible membrane internalization by budding of hydraulic blisters.

## Discussion

In this work, we study the hydraulic fracturing of lipid vesicles strongly adhered to other lipid membranes through mobile bonds. We show that the patterns of membrane blisters in our system and their evolution resemble the process of microlumen formation and coarsening in embryonic tissues[7]. Using our simplified experimental system in combination with theoretical modeling and numerical simulations, we are able to identify the physical principles controlling nucleation, spatial pattern, and dynamics of hydraulic cracks. We further identify the regions of parameter space leading to pattern formation characterized by an intermediate degree of confinement of the adhesion cleft. If too confined, osmotic imbalances cannot penetrate the interstice, whereas if insufficiently confined, water efflux can escape the cleft without compromising adhesion. Our work suggests a framework to estimate the poorly characterized transport parameters[19] (diffusivity, water, and membrane mobility) of nanoscale adhesion clefts from micron-scale hydraulic fracture patterns, which we show strongly depend on the properties and density of adhesion molecules. Our

observations further show that bond and lipid mobility depends not only on lateral crowding as widely appreciated, but also on the magnitude of the driving force, akin to jammed colloidal glasses[28]; close to equilibrium, lipids and bonds in the adhesion patch and the contact line appear arrested, whereas following the osmotic shock, lateral gradients of bond stretching generate a strong driving force to yield and fluidize the adhesion, enabling its remodeling. Arrested spreading results in lower membrane tension and ostensible fluctuations, providing very different experimental conditions as compared to the initial strong adhesion regime. We show that fluctuations favor bond breaking or patch shrinking depending on bond mobility, and that lower membrane tension leads to irreversible budding of blisters and membrane internalization, akin to precursors of endocytic vesicles. Previous observations of blister formation following osmotic shocks in substrate-adhered vesicles via non-specific physical interactions[39,40] suggest a broader generality of our results.

Our work provides a physical basis for reconfigurations of cell–cell adhesions. In general, biological patterning and reshaping during development results from an interplay between mechanics and biochemical regulation[41]. In the context of luminogenesis, our work identifies the physical rules enabling the initial patterning of profuse hydraulic cracks at every cell–cell junction, on top of which the previously identified mechanism guiding coarsening by gradients of cell surface tension can act to position the blastocoel[7]. For instance, our results suggest that rather than hydraulic confinement by tight junctions at the cell-medium interface, profuse cracking requires reduced water mobility throughout cell–cell adhesions in the embryo. Furthermore, healing of most adhesions requires avoiding irreversible budding of pockets by keeping cellular tension sufficiently high. In the context of adhesion remodeling and signaling by endocytosis, our work suggests that membrane internalization by budding of hydraulic blisters may constitute a physical pre-patterning mechanism for endocytic vesicles, subsequently tamed by known biochemical regulatory pathways[42,43]. Our mechanism of internalization may also provide a physical template for the clearance of interstitial water by macropinocytosis following the post-injury hydraulic fracturing in the skin of freshwater fish[4]. Thus, we establish a mechanism involving both interstitial pressure and tension for the mechanical regulation of endocytosis at cell–cell contacts[44].

## Methods

### Consumables
1,2-dioleoyl-sn-glycero-3-phosphocholine (DOPC), 1,2-dioleoyl-sn-glycero-3-phosphoethanolamine-N-(cap biotinyl) (sodium salt) (b-DOPE) and 1,2-dipalmitoyl-sn-glycero-3-phosphoethanolamine-N-(lissamine rhodamine B sulfonyl) (ammonium salt) (Rhod-DPPE) were all purchased from Avanti Polar Lipids (Alabaster, AL) and used without further purification. NeutrAvidin Protein, DyLight 488 (NAV) was purchased from Thermo Fisher Scientific. Chloroform, trizma hydrichloride (Tris · HCl), glucose, and sucrose were purchased from Sigma Aldrich. Microscope slides and cover glasses from VWR (catalog no. 48366 045) were used. For the preparation procedure of GUVs, we used Indium Tin oxide-coated glasses (ITO glasses) from Delta Technologies (no. X180). CholDNA_Short and BioDNA were purchased from Integrated DNA Technologies, while CholDNA_Long was purchased from Eurogentec.

### Substrate preparation and chamber
Glass cover slides were washed with isopropanol and ultrapure water (18.2 MΩ, 0.5 ppm organics, Merck Millipore), dried with nitrogen flow and further cleaned by exposing to air plasma, at a pressure of 1 mbar and a power of 300 watts (VacuLAB Plasma Treater, Tantec), for 40 s to render the cover-slide clean and hydrophilic. The experimental chamber was assembled by sticking a PDMS spacer onto the cleaned glass substrate. The total volume of the chamber was 500 µl.

### Preparation of DNA linkers
The biotinylated, double-cholesterol's DNA linkers comprised three synthetic DNA oligonucleotides. Two oligonucleotides were modified with a cholesterol-TEG (triethylene glycol) moiety (CholDNA_Long and CholDNA_Short), while the third (BioDNA) was labeled with a biotin modification. Sequences of the strands were as follows:

CholDNA_Long: 5′-CholTEG - CAA TCA CAC CAC AAA CAC CCA ACA CAA CAA CAA ACC-3′
CholDNA_Short: 5′-GTG TTT GTG GTG TGA TTG - CholTEG-3′
BioDNA: 5′-5BiosG - TTT GGT TTG TTG TTG TGT TGG-3′

Similar DNA constructs, with the exception of the biotin modification, were used in ref. [45]. The three oligonucleotides were designed to assemble as shown in Fig. S1. The presence of three unpaired thymine between the double-stranded DNA segment and the biotin modification imparts flexibility. The constructs were formed by combining all three constructs in stoichiometric ratios, at a concentration of 10 µM, in TE buffer + 100 mM NaCl. Samples were then heated up to 96 °C and let cool down to 20 °C over 4 h on a thermal cycler. Correct assembly was verified using gel electrophoresis (Fig. S1), for which modified versions of the CholDNA_Long and CholDNA_Short strands were used, lacking the cholesterol modifications to prevent micellization.

### Preparation of supported lipid bilayer (SLB)
SLB were prepared using vesicle fusion. Briefly, a thin film of 2mg lipids formed by 99.5; 98.7 or 95.7 mol% DOPC with 0.2, 1 or 4 mol% b-DOPE, respectively, and 0.3 mol% Rhod-DPPE were dried in a vacuum dessicator overnight on the wall of glass vial. The dried lipid film is rehydrated in lipid buffer (10mM Trizma base; 150 mM NaCl and 2mM CaCl$_2$, pH ≈ 7.5) to a final concentration 1mg/mL. The resulting suspension is then sonicated using a tip sonicator operated in a pulsed mode at 20% power for 10 min with refrigeration to generate small unilamellar vesicles (SUV's) from the lipids. The solution is then centrifuged at $100 \times g$ for 10 mn in an Eppendorf centrifuge to remove titanium particles. SUV suspensions were stored at 4 °C under nitrogen and used within a week. A dilution of the SUVs suspension with lipid buffer at a 1: 4 volume ratio is spread over the clean hydrophilic glass cover-slide in a final 200 µl volume created by the PDMS chamber (see above). Incubation for about 30–60 min results in the formation of a supported lipid bilayer. The SLB was then thoroughly washed with glucose solution having a concentration of 300 mM (isotonic relatively to the sucrose solution in which the GUVs have been prepared). This is done to remove unfused SUV's and the lipid buffer.

### Preparation of giant unilamellar vesicles (GUV)
GUVs with the same composition as that of the SLB were produced via electroswelling[46]. Briefly, 50 µl of the solution containing the lipid mixture was dispersed on two titanium oxide-coated glass slides and dried in a vacuum desiccator. The lipid-coated ITO slides facing each other are put together with a Teflon spacer to form a capacitor cell filled with 300 mM sucrose solution, and connected to an alternating electric field at 10 Hz and 2V peak to peak amplitude overnight. The GUV's were then extracted from the chamber, stored in an Eppendorf vial and used within 2–3 days.

### Immobilization of giant unilamellar vesicles
To bind GUVs to SLB with biotin-Neutravidin bonds (b-NAV), the GUVs and SLB were prepared from the same lipid stock solutions of DOPC and b-DOPE as described above. Before vesicle adhesion, the SLB was incubated with an excess of NAV at a final concentration of 60 µg/ml for 30 mn and then rinsed with glucose 300 mM solution to remove excess protein. Following that, 2–5 µl of GUV solution was added to the chamber and incubated for 30 mn to allow the GUVs to sediment and adhere onto the SLB. The solution is then washed with 300 mM glucose solution to remove unbound vesicles.

The b-NAV GUV-GUV system were formed in a comparable manner to the SLB-GUV except that vesicles were added at higher concentration to the SLB, at the same time with the NAV solution to facilitate binding GUV-SLB and GUV-GUV binding.

To prepare the DNA-b-NAV systems with DNA spacers, we followed the experimental procedure described in Amjad et al.[47]. The DNA constructs were stored in DNA buffer at a concentration of 5 μM. SLBs and GUVs were prepared from a lipid mixture of 99.7 mol% DOPC and 0.3% mol Rhod-DPPE as above. The SLBs were first rinsed with DNA buffer (300 mM) (Tris EDTA (1X); 100 mM NaCl and 87 mM glucose), after which a 0.5 μl of the GUV solution and a certain amount of the DNA constructs solution were added to it, to achieve between 1 to 8 mol% DNA-b linker density in both SLB and GUVs. After 1 hour incubation, a certain amount of the NAV solution was added to achieve a ratio of NAV/DNA-b of 1/4 to allow each NAV to bind to four b-DNA ligands. In reality, however, the DNA membrane density is expected to be lower than this due to non-specific binding of DNA to the sides of the chamber. Moreover, it is unlikely to reach DNA density of even 4 mol% due to strong steric/Coloumb repulsion between the DNA rods.

## Osmotic shocks

By the time GUVs were bound to the SLB, all samples had a final volume of 400 μL. To subject the b-NAV GUVs to hyper-osmotic shocks of 25, 50, 100, and 150 mM osmotic shocks, half of the volume of the chamber (200 μL of the 300 mM osmolarity) was replaced by glucose solutions of 350, 400, 500, and 600 mM, respectively. For the DNA-b-NAV GUVs, the shock solutions were 350 mM (Tris EDTA (1X) + 100 mM NaCl + 137 mM glucose), 400 mM (Tris EDTA (1X) + 100 mM NaCl + 187 mM glucose); 500 mM (Tris EDTA (1X) + 100 mM NaCl + 287 mM glucose) and 600 mM (Tris EDTA (1X) + 100 mM NaCl + 387 mM glucose), respectively.

The precise osmolarity of the shock solutions was measured for each experiment with an osmometer (Osmomat 3000, Gonotec GmbH, Berlin, Germany). After the addition of the shock solution, the chamber was covered to prevent further osmolarity changes due to evaporation.

## Imaging and analysis

The imaging of the adhesion zone between the SLB and GUV throughout the osmotic shock was performed with an inverted optical microscope Nikon Eclipse Ti-E and a 60x numerical aperture, oil immersion objective in combination with an Andor camera Neo 5.5 sCMOS (Oxford Instruments). The integrated perfect focusing system (PFS) in the microscope allows us to follow automatically the surface which change its focal plane during the application of the osmotic shock. The open source image processing package FIJI was used for the image analysis. The changes in the adhesion area and intensity in response to the osmotic shock are performed by first subtracting the background of the fluorescence images and then applying an appropriate thresholding to generate a binary stack from whch we extract the total patch area and the intimate adhesion area.

Confocal images were acquired using a Zeiss LSM 880 Fast AiryScan and a Plan-Apochromat 63x numerical aperture 1.4 Oil immersion objective. The three-dimensional (3D) reconstruction using the confocal stack was done using a Fiji plugin (ClearVolume)[48].

## Simulations

The equations describing the time-evolution of the system involve a set of fields $(z, \mathbf{v}, P, \Pi, \sigma)$ in the patch $\mathcal{D}(t)$ and the variables $(R, \theta, P_i, \Pi_i, \sigma_v)$ representing the state of the vesicle (Supplementary Note 2). We integrate these equations in time in a staggered way, by first solving the equations for $(z, \mathbf{v}, P, \Pi, \sigma)$ with a backward Euler approximation assuming fixed values of $(R, \theta, P_i, \Pi_i, \sigma_v)$ and then solving for the vesicle variables assuming fixed values for $(z, \mathbf{v}, P, \Pi, \sigma)$. To

discretize $(z, \mathbf{v}, P, \Pi, \sigma)$ in $\mathcal{D}(t)$ we consider a triangular mesh and use a second-order Lagrangian interpolation for $(z, \mathbf{V}, P, \Pi)$ and a first-order Lagrangian interpolation for $\sigma$ where here $\mathbf{V} = \mathbf{v} + v_n \mathbf{N}$ is the three-dimensional velocity of lipids and $\mathbf{N}$ is the unit normal to the membrane surface. To recover $z$ from $\mathbf{V}$, we note that since $\partial_t z = v_n$, we can approximate $z(t + \Delta t) \approx z(t) + (\mathbf{V} \cdot \mathbf{N})\Delta t$. To compute the tangential velocity $\mathbf{v}$, we project $\mathbf{V}$ onto $\Gamma_t$. To solve the balance of forces on the membrane we follow the procedure detailed in ref. 24. The equations are then solved using a finite element method implemented in hiperlife[25].

## Reporting summary

Further information on research design is available in the Nature Portfolio Reporting Summary linked to this article.

## Data availability

The data that supports the findings of this study can be found in the manuscript, its Supplementary Information, and the provided Source data file. Unprocessed microscopic images are available in the Durham University research data repository with identifier https://doi.org/10.15128/r2000000048.

## Code availability

The computer code used to perform all the simulations of this study as well as input files are available at https://doi.org/10.5281/zenodo.8277743.

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

## Acknowledgements

M.S. and C.D. acknowledge funding from Engineering and Physical Sciences Research Council (EPSRC Grant EP/P024092/1) and C.D. acknowledges funding granted by Durham University and the European Union's Seventh Framework Programme for research, technological development and demonstration under grant agreement no 267209. A.T-S. and M.A. acknowledge the support of the European Research Council (ERC-CoG No 681434). M.A. acknowledges the European Commission (H2020-FETPROACT-01-2016-731957), of the Spanish Ministry for Science and Innovation (PID2019-110949GB-I00) and of the Generalitat de Catalunya (ICREA Academia prize for excellence in research). IBEC and CIMNE are recipients of a Severo Ochoa Award of Excellence. L.D.M. acknowledges funding from a Royal Society University Research Fellowship (UF160152, URF21009) and from the European Research Council (ERC-STG No 851667 - NANOCELL). L.D.M. and R.L. acknowledge support from the Wiener-Anspach Foundation. For the purpose of open access, the authors have applied a Creative Commons Attribution (CC BY) licence to any Author Accepted Manuscript version arising.

## Author contributions

M.S., M.A., C.D., and A.T-S. designed research; C.D. and M.S. performed experiments; R.L. and L.D.M. provided DNA constructs and advised on DNA experiments; A.T.-S. and M.A. performed the modeling; C.D., A.T.-S., M.A., and M.S. analyzed the data, interpreted the results, and wrote the manuscript.

## Competing interests

The authors declare no competing interests.

 
