## [Peer Review File · Nature Communications]

REVIEWER COMMENTS

Reviewer #1 (Remarks to the Author):

This manuscript shows a beautiful series of minimal in vitro experiments meant to capture the physical mechanisms underlying blisters in cell-cell adhesions.

The experiments are highly systematic. The authors vary the osmotic shock (i.e. driving force for blistering), the adhesion strength, and spacing between the bilayer (to vary the permeability). One particularly compelling aspect of this work is that the more quantitative results are compared and connected to more qualitative observations of vesicle-vesicle contacts.

In addition to these powerful experiments, the authors develop a minimal physical model that nicely captures the essential experimental features of early blister formation. Admirably, the authors challenge this model, and demonstrate how where these idealization break down through a number of mechanisms, including the budding of blisters.

I think that this is a very nice contribution to the physics of adhesive lipid bilayers with clear implications for biological systems.

I think this manuscript is a great fit for Nature Communications and I would like the authors to make a few modifications to clarify the essential ideas.

Line 53: I don't think that "young-dupre model" tells the reader much, and is a bit confusing, as Young and Dupre never worked on adherent vesicles. Instead of a name, offer an explanation. Maybe you could say "If vesicles behave like droplets, we expect them to spread when their tension drops" or something else descriptive, like that.

Line 32, 203, 229: The term 'molecular crowding' usually refers to high densities of macromolecules in solution, which primary manifest through elevated osmotic pressure. Therefore, I was confused by the authors' use of crowding here, which simply refers to the small gap between the vesicles (as far as I can tell). I think that the authors mean to use the word 'confinement', which has an impact on motility (and permeability).

I don't think the mechanism for the tension drop at late times if very clearly explained. As far as I understand it, the basic idea is that once the edges of the vesicle come in close contact with the SLB, the osmotic pressure is only equalize by transport across the membrane, not through the gap, right? This leads to familiar vesicle deflation. Do I get this right? I think the discussion around this point (around lines 230) could be more clear and direct. Even then, it's still not completely obvious that reduced tension should lead to budding instead of spreading. The authors cite some papers, but, again, could they offer a sentence or two of physical explanation?

Reviewer #2 (Remarks to the Author):

In their manuscript, Dinet, Torres-Sánchez and colleagues explore the hydraulic fracturing of adhesive membranes using biomimetic experiments, theoretical modeling and simulations. They carefully dissect multiple physical parameters involved in the pattern of formation of blisters inflating between adhesive sites and how they coalesce. They compare the predictions of their simulations with experimental manipulations to explore the contributions of adhesive bond density, adhesive bond length and osmotic pressure. Both experiments and theory provide unique insights into a recently described biological process, which remains poorly understood. This appears to me as extremely powerful and promising approaches. Some of the conclusions reached by the authors seem intuitive, yet important, and others are extremely interesting. This will surely inspire exciting new experiments and theories.

The main text is clear and well written, the figures are elegant and mostly self-explanatory. The authors have decided to keep most of the theory in several dedicated sections. It is still unclear to me how they wish to arrange them in the published manuscript but having different elements in too many different places may affect the ability of readers to find information.

I list below a few questions, which may need clarification in a revised manuscript.

1 – During osmotic shock, the volume of the GUV is decreasing. This is not really mentioned explicitly throughout the manuscript. How much is the GUV volume changing during osmotic shock? How critical is the GUV volume change for the theory? Is it taken into consideration? In particular, how critical is GUV volume change to generate excess membrane that flows into the blisters?

2 – Before osmotic shocks, is the distance between the GUV and substrate or GUV-GUV homogeneous throughout the contact or is the distance shorter in the center compared to the periphery?

Interestingly, the authors vary the bond length experimentally, which impacts the pattern of fracturing and coarsening, via changes in osmolyte and water mobilities according to the authors conclusions. Could the authors also vary the bond length (z_0) in their simulation? Does that also result in differences in osmolyte and water mobilities? Are there other effects?

3 - Over long time scales, blisters disappear (F4d, FS3). When are the fractured area measured in F4a and b? This information is essential.

Also, the authors mention that, after coarsening, there are no more blister “in analogy to the healing of hydraulically fractured tissues”. I am not sure how much the analogy holds. In tissues, coarsening is actively controlled by cell contractility (Casares et al 2015, Dumortier et al 2019) or other active processes (Kennard et al 2022). Here, the process is passive and the GUV may have recovered its pristine adhesive patch but it seems now to be much larger than before the shock. This statement may be amended or clarified.

4 - When using longer adhesive bonds, the authors mention enhanced Ostwald ripening. What do they mean by that precisely? Is that something that they measured?

5 - Movies of experiments are spectacular but need spatial and temporal information.

6 - There seems to be issues with the colors of F2b. The purple of membrane tension is very close to the red of hydraulic pressure.

Point-by-point response to reviewers and statement of changes

Reviewer #1

This manuscript shows a beautiful series of minimal in vitro experiments meant to capture the physical mechanisms underlying blisters in cell-cell adhesions. The experiments are highly systematic. The authors vary the osmotic shock (i.e. driving force for blistering), the adhesion strength, and spacing between the bilayer (to vary the permeability). One particularly compelling aspect of this work is that the more quantitative results are compared and connected to more qualitative observations of vesicle-vesicle contacts. In addition to these powerful experiments, the authors develop a minimal physical model that nicely captures the essential experimental features of early blister formation. Admirably, the authors challenge this model, and demonstrate how where these idealization break down through a number of mechanisms, including the budding of blisters. I think that this is a very nice contribution to the physics of adhesive lipid bilayers with clear implications for biological systems. I think this manuscript is a great fit for Nature Communications and I would like the authors to make a few modifications to clarify the essential ideas.

We thank the reviewer for these positive comments.

Line 53: I don't think that "young-dupre model" tells the reader much, and is a bit confusing, as Young and Dupre never worked on adherent vesicles. Instead of a name, offer an explanation. Maybe you could say "If vesicles behave like droplets, we expect them to spread when their tension drops" or something else descriptive, like that.

Following the reviewer's suggestion, we have re-phrased this sentence without mentioning the Young-Dupré model to prevent confusion (paragraph starting at line 55).

Line 32, 203, 229: The term 'molecular crowding' usually refers to high densities of macromolecules in solution, which primary manifest through elevated osmotic pressure. Therefore, I was confused by the authors' use of crowding here, which simply refers to the small gap between the vesicles (as far as I can tell). I think that the authors mean to use the word 'confinement', which has an impact on motility (and permeability).

The term "crowding" has also been applied to proteins attached to lipid bilayers referring to high lateral packing in the surface, e.g. in the context of adhesion [<https://doi.org/10.1016/j.bbamcr.2015.05.025>] or membrane reshaping [<https://doi.org/10.1038/ncb2561>], but we agree with the referee that this point requires clarification. In all instances where it was used, we have modified the wording to be more specific and clear.

For instance, the text in line 32 now reads "At higher bond density, the lateral crowding of the NAVs confined within the thin (5-6 nm) interstitial space", where it is clear that we specifically refer to the lateral crowding.

In line 118-119, we have replaced "..., hence explaining why bonds constrained by crowding and pinning appear immobilised during FRAP, but mobile during fracturing" by "..., hence explaining why bonds with reduced mobility at high concentrations appear immobilised during FRAP, but mobile during fracturing".

One key point in the manuscript is that "crowding" leads to a solid-like behavior under FRAP but a fluid-like behavior enabling adhesion remodeling during hydraulic fracture, when driving forces are

much larger. To specifically highlight this idea and avoid lumping it into the concept of “crowding”, we have added the following text in line 121:

“Adhesion patches with high bond density behave similarly to dynamically arrested dense colloidal glasses, which attain fluid-like behaviour only beyond a certain yield stress^{28;29}.”

We think that this analogy with dense colloidal glasses will be clarifying to many readers. This also allows us to refer to this behaviour more easily, e.g. as “arrested bond mobility”.

In line 209, we have replaced “and reduce molecular crowding” by “which reduces the volume fraction of the linker complexes in the interstitial gap”.

In line 250, we have replaced “In the absence of strong ‘fracturing’ forces, the bonds and lipids become immobilised again by crowding and substrate effects...” by “Right after the osmotic shock, fracturing forces can move bonds. However as the system approaches osmotic equilibrium, bonds become dynamically arrested”.

In line 263, we have added the adverb “laterally” to specify what we mean by crowding.

We think that the revised manuscript addresses the concern of the referee and clarifies the text for all readers.

I don't think the mechanism for the tension drop at late times is very clearly explained. As far as I understand it, the basic idea is that once the edges of the vesicle come in close contact with the SLB, the osmotic pressure is only equalized by transport across the membrane, not through the gap, right? This leads to familiar vesicle deflation. Do I get this right? I think the discussion around this point (around lines 230) could be more clear and direct. Even then, it's still not completely obvious that reduced tension should lead to budding instead of spreading. The authors cite some papers, but, again, could they offer a sentence or two of physical explanation?

Following the recommendation of the referee, we have completely rewritten the paragraph starting in line 242. We think that the new explanation about the mechanisms leading to the tension drop is much clearer and explicit in the revised version.

Reviewer #2

In their manuscript, Dinet, Torres-Sánchez and colleagues explore the hydraulic fracturing of adhesive membranes using biomimetic experiments, theoretical modeling and simulations. They carefully dissect multiple physical parameters involved in the pattern of formation of blisters inflating between adhesive sites and how they coalesce. They compare the predictions of their simulations with experimental manipulations to explore the contributions of adhesive bond density, adhesive bond length and osmotic pressure. Both experiments and theory provide unique insights into a recently described biological process, which remains poorly understood. This appears to me as extremely powerful and promising approaches. Some of the conclusions reached by the authors seem intuitive, yet important, and others are extremely interesting. This will surely inspire exciting new experiments and theories. The main text is clear and well written, the figures are elegant and mostly self-explanatory.

We thank the reviewer for these positive comments.

1 – The authors have decided to keep most of the theory in several dedicated sections. It is still unclear to me how they wish to arrange them in the published manuscript but having different elements in too many different places may affect the ability of readers to find information.

In our original submission, the theory is presented in three different places, (1) as a narrative in the main text, so that a wide diversity of readers can understand the main ideas, (2) as a Supplementary Note, where the equations are derived from balance laws and constitutive relations and the assumptions are precisely discussed and (3) as a one-page self-standing summary of the theory, which enumerates the unknowns, provides all governing equations, initial and boundary conditions, and lists the model parameters.

We think that these three ways to present the theory may help different readers. For instance, a reader interested in the theory may want to check the summary in the box, and only read the supplement to understand the rationale behind a specific equation. However, we agree with the referee that readers need guidance to know what information is where. In the revised manuscript, we have added a sentence in line 96 for this purpose.

The theory box could be either a display item in the main text (our preferred option) or may be placed within the Supplementary Note 2, which is what we have done in our revised supplement. We are open to suggestions by referees and the editor.

2 – During osmotic shock, the volume of the GUV is decreasing. This is not really mentioned explicitly throughout the manuscript. How much is the GUV volume changing during osmotic shock? How critical is the GUV volume change for the theory? Is it taken into consideration? In particular, how critical is GUV volume change to generate excess membrane that flows into the blisters?

We have now added an estimation for the volume loss of the vesicle shown in Figure 1a (line 43 and Supplementary Note 1), and have discussed the link between the volume loss, excess surface area and formation of blisters (paragraph starting in line 55).

The theory accounts for volume changes, since both the shape of the free-standing part of the vesicle and that within the adhesion patch are dynamical variables of the model, and these variables define the vesicle surface and the volume it encloses. We have clarified this in line 111. Our calculations show that the volume change depends essentially on the strength of the osmotic shock and is quite insensitive to any other parameter, as expected in the typical high osmotic strength regime of biological solutions (this is discussed now in Supplementary Note 3, see also the figure below). In experiments the osmotic shock ranges from 25 mM to 150 mM. In those of Fig. 1, it is of 100 mM, which corresponds to a change of 250 kPa in osmotic pressure, and in agreement with the model calculation the volume change is of 20%. In the simulations reported in the paper, we consider 50 mM, which corresponds to 125 kPa, and find volume changes close to 15%.

The volume change is an essential consequence of the osmotic shock in our system, which drives all out-of-equilibrium phenomena that we report. It is essential to drive water by permeation into the interstitial space as we report in Fig. 2(b,c) and in movie S3, but most of the volume loss goes into the surrounding medium through the free-standing part of the vesicle. If water permeation was only possible in the adhered part (by confinement to an impermeable medium), then a much smaller volume loss would be enough to drive blistering. One could also envision a situation in which blisters draw area from the vesicle at fixed vesicle volume, e.g. if water was driven into the interstice from a poroelastic substrate (this was done in Casares et al, 2015 and in Kosmalka et al, 2015). However, these scenarios are not pertinent to our experiments.

3 – Before osmotic shocks, is the distance between the GUV and substrate or GUV-GUV homogeneous throughout the contact or is the distance shorter in the center compared to the periphery?

The distance between the GUV and SLB appears homogeneous, down to a resolution of 1 nm, as demonstrated by previous Reflection Interference Contrast Microscopy images (see for example Fenz et al. Langmuir 2009, 25, 1074-1085), as well as by own images for verification (data not shown in the paper).

4. Interestingly, the authors vary the bond length experimentally, which impacts the pattern of fracturing and coarsening, via changes in osmolyte and water mobilities according to the authors conclusions. Could the authors also vary the bond length (z_0) in their simulation? Does that also result in differences in osmolyte and water mobilities? Are there other effects?

We thank the reviewer for this suggestion. Accordingly, we have performed new simulations (Movie S5 and main text in lines 215 to 220) representative of the DNA model, which capture the changes observed experimentally. In these simulations, we increase z_0 , but also the osmolyte diffusivity and water mobility. We discuss the rationale for these parameters in Supplementary Note 3.

5 - Over long time scales, blisters disappear (F4d, FS3). When are the fractured area measured in F4a and b? This information is essential.

In the revised manuscript, we have explicitly stated in the captions of Figures 4 and S4 that that fractured area is measured when it is maximum.

6. Also, the authors mention that, after coarsening, there are no more blister “in analogy to the healing of hydraulically fractured tissues”. I am not sure how much the analogy holds. In tissues, coarsening is actively controlled by cell contractility (Casares et al 2015, Dumortier et al 2019) or other active processes (Kennard et al 2022). Here, the process is passive and the GUV may have recovered its pristine adhesive patch but it seems now to be much larger than before the shock. This statement may be amended or clarified.

We agree with the referee that the mechanism of blister coarsening is actively driven in cells, and entirely passive in our system. We also agree that our system does not recover the exact initial state because of spreading. However, both in Casares/Dumortier and in our in vitro system, healing is the result of an increase in the “tension-pressure” ratio; In the cellular systems, this is due presumably to an active increase in cell membrane tension while in our system it is due to passive decrease of pressure as it approaches equilibrium. Furthermore, both in cells and in our systems healing requires redistribution of water from small pockets to larger pockets, the blastocoel or the external medium. Following the comment of the referee, we have reworded this statement and specified the limits of the analogy (lines 224-225).

We also thank the referee for bringing up the work by Kennard et al, 2022, which is very pertinent to our study but we had overlooked in our original manuscript. In the revised manuscript, we have

cited this paper alongside with Casares et al in the introduction, and added a new sentence in the conclusions, line 307, where we speculate about the relation between the irreversible internalization of buds in our system and the clearance by macropinocytosis of interstitial water following hydraulic fracture in zebrafish.

7 - When using longer adhesive bonds, the authors mention enhanced Ostwald ripening. What do they mean by that precisely? Is that something that they measured?

We have clarified in the manuscript that enhanced Ostwald ripening refers to the quicker loss of fractured area. This is now supported by additional figure in SI (Figure S7), which compares the decrease of A_{fracture} for b-NAV samples at 4mol% and for DNA-b-NA samples. The decrease in A_{fracture} proceeds without visible blister budding and leads to recovery of pristine adhesion. It is associated with faster water Darcy mobility and manifests in the discharge of blisters into larger ones, or in the external medium. Our new Video S5 compares simulations representative of b-NAV and DNA-b-NA systems, which show enhanced coarsening by Ostwald ripening in the latter.

8 - Movies of experiments are spectacular but need spatial and temporal information.

We thank the reviewer for pointing this out, which has been corrected in this resubmission.

9 - There seems to be issues with the colors of F2b. The purple of membrane tension is very close to the red of hydraulic pressure.

We have now changed the colormap for membrane tension slightly so that it can be easily distinguished from the hydraulic pressure. We have made the same change in Video S2.

Statement of changes

All changes in the main text are highlighted in colour. We list below the changes made to the supplementary information.

- Additional Supplementary Figure 7 to substantiate our answer to question 7 of reviewer #2.
- Additional Supplementary Video of simulation with longer linkers representative of the DNA system to address question 4 of reviewer #2.
- Additional Supplementary Video of very long simulation achieving complete recovery of the adhesion patch. To achieve this, we had to reduce the system size.
- Addition of time-stamp and scalebar to experimental movies.
- We have replaced the original Supplementary Video S3 by a different one of one of the new simulations, which better shows water and membrane flows required to enable the dynamics of blisters.
- An additional sentence at the end of Section “Geometry of adhered vesicles” with a quantification of the volume changes following osmotic shock to address question 2 of reviewer #2.
- In Supplementary Note 3, we have added a new section “Initial contact angle” to detail our choices and their effect, and we have detailed and justified our choices for the bond length, diffusivity and Darcy mobility corresponding to the new simulations of the DNA system, responding to question 4 of reviewer #2.
- We have moved the Theory Box to the end of “Supplementary Note 2: Theoretical model” following question 1 of reviewer #2.

REVIEWERS' COMMENTS

Reviewer #1 (Remarks to the Author):

The authors have very thoughtfully responded to my suggestions and I have no hesitation to support the publication of the manuscript.

Reviewer #2 (Remarks to the Author):

The authors have addressed all of my comments. Congratulations to them.

Minor suggestion:

Timestamps in dark red in some movies are difficult to see.